# The effectiveness of high-intensity interval training versus moderate intensity continuous training in prehabilitation among patients undergoing major abdominal surgery: A study protocol

Suriah Ahmad[1,2], Nor Azura Azmi[1]*, Nur Ayub Md Ali[3,4], Md Ali Katijjahbe[2,3], Ian Chik[4], Syarifah Noor Nazihah Sayed Masri[5], Adzim Poh Yuen Wen[4]

1 Physiotherapy Programme, Center for Rehabilitation and Special Needs Studies, Faculty of Health Science, Universiti Kebangsaan Malaysia, Kuala Lumpur, Malaysia, 2 Physiotherapy Unit, Hospital Canselor Tuanku Muhriz, Universiti Kebangsaan Malaysia, Kuala Lumpur, Malaysia, 3 Heart and Lung Centre, Hospital Canselor Tuanku Mukhriz, Universiti Kebangsaan Malaysia, Kuala Lumpur, Malaysia, 4 Surgery Department, Faculty of Medicine, Universiti Kebangsaan Malaysia, Kuala Lumpur, Malaysia, 5 Department of Anaesthesiology and Intensive Care, Hospital Canselor Tuanku Muhriz, Kuala Lumpur, Malaysia

* nazura.azmi@ukm.edu.my

## Abstract

### Background

Prehabilitation programmes for major abdominal surgery enhance patients' condition preoperative and promote recovery by building surgical resilience. However, the precise protocol of prehabilitation pertaining to the prescription of exercise training remains undefined. This paper describes the protocol of the study that aims to evaluate the effectiveness of high-intensity interval training (HIIT) compared to moderate intensity continuous training (MICT) among patients undergoing major abdominal surgery.

### Methods

This is pragmatic double-blind randomized controlled trial, with parallel group, concealed allocation and blinding of patients and assessors. A total of 70 participants will be recruited from the surgery and anaesthetic clinic at the Hospital Canselor Tuanku Muhriz. Participants will be randomly allocated 1:1 to either receive HIIT (intervention group) or MICT (control group) with 35 participants in each group. Both groups will receive body conditioning and respiratory muscle strength exercises of HIIT for participants in the intervention group, while MICT for the control group. This will be one-hour therapist-supervised intervention sessions for at least 4 weeks duration with 1–3 sessions per week before the surgery. The patient will be assessed at baseline, before the operation at 4-week following intervention, prior to discharge, at 4 weeks

**Data availability statement:** No datasets were generated or analysed during the current study. All relevant data from this study will be made available upon study completion.

**Funding:** The author(s) received no specific funding for this work.

**Competing interests:** The authors have declared that no competing interests exist.

and 3 months postoperatively. The primary outcome measures are 6-minute walking test and maximum inspiratory pressure. The secondary outcomes will be multidomain recovery of physical performances, psychological, and quality of life. All data will be analyzed using descriptive and inferential statistics, particularly Mixed Model ANOVA. The statistical significance value will be set at $p < 0.05$. The trial is currently recruiting participants.

## Conclusions

The findings of this research will inform guidelines on optimal exercise dosage and intensity for prehabilitation in patients undergoing major abdominal surgery.

## Protocol registration

The protocol of this study is registered in the Australia New Zealand Clinical Trial Registry with registration number ACTRN12625000023459.

---

## Introduction

Abdominal surgery is the most common surgery performed worldwide [1]. Elective major abdominal surgery (MAS) is defined as a planned of any abdominal surgery where the total incision length is ≥ 5 cm involving open, laparoscopic, or minimally invasive procedures performed under general anaesthesia, with an incision into the abdominal cavity, visceral manipulation, and at least an overnight hospital stay [1].

The most common complication following MAS is a postoperative pulmonary complication (PPCs) with a reported incidence of 13–53% [2]. Most PPCs manifest within the initial 3 days following surgery [3,4]. Evidence indicates that PPCs are associated with prolonged hospital stays [5,6], an elevated risk of unplanned readmissions [5], diminished quality of life and physical function [7], and an increased risk of all-cause mortality within 12 months [2,5]. Owing to the high incidence of PPCs and their substantial impact on patients, the prediction and targeted prevention of PPCs are strongly recommended [8].

Enhanced Recovery after Surgery (ERAS) is an evidence-based preoperative, intraoperative and postoperative multi-model care pathway that aims to reduce stress response to surgery and accelerate post-op recovery through enhancing patient mobilization, reducing complication rates after surgery, decreasing hospital length of stay and reducing healthcare costs [1,9,10]. Since ERAS was first implemented within hospitals over twenty years ago, post-surgical outcomes have improved for patients [11]. Length of stay has decreased, with no subsequent increase in readmission rates [12], with concurrent improvements in clinical outcomes whilst having a beneficial impact on healthcare resources. ERAS originated in elective colorectal surgery but has spread to other surgical subspecialties, including, but not limited to, gastrointestinal, hepatobiliary, orthopaedic, cardiac, thoracic, head and neck, breast and gynaecologic surgery [13]. The ERAS approach comprises preoperative, intra-operative, and postoperative components, with optimization of the patient's physical

status before surgery as a key focus [9]. Therefore, prehabilitation program by physiotherapists is essential in enhancing patients' recovery and improving their outcomes after surgery including physical performances, psychological and health related quality of life [2,14,15].

The role of physiotherapy within ERAS pathways is important in both preoperative and postoperative routines [16]. Implementing a preoperative strength programme has been shown to promote musculoskeletal improvements in preparation for a forthcoming physiological stressor, and is an emerging key component of ERAS [16]. A literature review found preoperative exercise in patients scheduled for cardiovascular, thoracic, abdominal and major joint replacement surgery to be well-tolerated and effective [17]. Postoperative exercise programmes are also recommended by ERAS guidelines, promoting muscle hypertrophy and the return to function after major surgery [18].

The standard prehabilitation program may include aerobic exercise which often moderate-intensity continuous training (MICT) is used to improve cardiovascular fitness [19], resistance training to strengthen muscles and improve overall physical function [16], breathing exercise to enhance lung capacity and respiratory muscle strength [20], and education with lifestyle counselling includes guidance on smoking cessation, nutrition, and overall physical activity [21]. High-intensity interval training (HIIT) is a bolus-dosing approach that efficiently increases cardiorespiratory fitness (CRF) and is feasible in most surgical populations. High-intensity interval training involves repeated aerobic high-intensity intervals at approximately 80% of the maximum heart rate, followed by active recovery [22–24]. The rapid increases in CRF elicited with HIIT is appealing for preoperative patients, and in the context of pathology, age, and comorbidities, the volume of training stimulus required to improve CRF can often be achieved [24].

HIIT provides advantages in terms of time efficiency, greater cardiorespiratory fitness, improved metabolic health and more effectiveness in preserving muscle mass. Studies demonstrated that HIIT, required significantly less time than MICT, produced similar cardiovascular and metabolic health improvements [25–27]. The study highlighted the time efficiency of HIIT as a major advantage for patients with limited time before surgery. The study by Weston et al. (2014) showed that HIIT led to greater improvements in $VO_2$ max compared to MICT in patients with cardiovascular disease [28]. Given that $VO_2$ max is a key indicator of cardiorespiratory fitness, HIIT's superior outcomes highlight its potential benefit in prehabilitation [28].

The current literature highlights the significant impact of prehabilitation on improving patient outcomes in the context of upper abdominal surgery. Prehabilitation programs incorporating aerobic, resistance, and breathing exercises, alongside education and lifestyle counselling, have been shown to enhance cardiorespiratory fitness, reduce postoperative complications, and expedite recovery. While moderate-intensity continuous training (MICT) is commonly used in such programs, high-intensity interval training (HIIT) has emerged as a time-efficient and potentially superior alternative, particularly in improving $VO_2$ max and preserving muscle mass.

Despite these findings, there is a lack of standardized exercise protocols tailored to the prehabilitation phase for patients undergoing major abdominal surgery. Additionally, limited data exist on the health status and functional outcomes of these patients throughout the surgical timeline. This gap warrants further investigation into the comparative effectiveness of HIIT and MICT, particularly in terms of their impact on multidomain recovery. By addressing these research gaps, this study protocol aims to provide valuable insights into optimizing prehabilitation strategies, ultimately improving the quality of care and recovery outcomes for patients undergoing major abdominal surgery.

Therefore, this study intended to evaluate the effectiveness of HIIT over MICT prehabilitation training on cardiorespiratory fitness and maximum inspiratory pressure among patients undergoing major abdominal surgery. This study also aims to investigate the effectiveness of HIIT versus MICT prehabilitation training on physical performances, upper limb strength, body composition, fatigue, psychological and health-related quality of life. We hypothesize that a 4-week HIIT program during prehabilitation is non-inferior to MICT in improving the targeted outcomes following major abdominal surgery, meaning the difference between the two interventions does not exceed the pre-specified non-inferiority margin (Δ).

## Methods

### Study design

This is a phase II, prospective, two-arm parallel group, assessor and participants blinded randomized controlled trial (RCT) that will be carried out at a single tertiary care hospital in Malaysia. Participants will be randomly allocated 1:1 to either control group of MICT or the intervention group of HIIT. The study was approved by the local Research Ethics Boards of the participating site, Research Ethical Committee of Universiti Kebangsaan Malaysia (UKM: JEP-2024–527), ensuring compliance to the ethical principles of the Helsinki Declaration.

The trial protocol has been registered with the Australian New Zealand Clinical Trials Registry (ANZCTR) under registration number ACTRN12625000023459. Any modifications to the study protocol, along with their rationale, will be reported to the ethics committee and updated in the trial registry accordingly. The method is reported in accordance with the Standard Protocol Items: Recommendations for Interventional Trials guidelines for clinical trials, [29] and the Template for Intervention Description and Replication reporting of interventions [30]. Fig 1 shows a schematic outline concerning enrolment, allocation, intervention, and assessment through the study.

### Study setting and samples

This study is will be conducted at a teaching hospital of the Hospital Canselor Tuanku Muhriz (HCTM), Malaysia. Participants will be recruited from the surgery and anaesthetic clinic at the HCTM. Consecutive eligible patients scheduled for elective major abdominal surgery between February 2025 and July 2026 will be prospectively invited to participate. Fig 2 summarises the design of the trial, and each of the trial's aspect is described in detail below.

### Eligibility criteria

Participants will be randomised to participate in the trial if they meet the eligibility criteria, give written informed consent and have completed baseline measurement testing performed by a blinded assessor. Participants will be informed that they will be randomised to receive either high-intensity interval training (HIIT) or moderate-intensity continuous training (MICT) preoperatively and will be allocated to one of two groups: (1) the intervention group (HIIT), or (2) the control group (MICT).

**Inclusion and exclusion criteria for participants.** Adults with the following inclusion criteria are eligible to participate: (1) aged 18 years old and over, (2) scheduled for elective major abdominal surgery, (3) able to perform the 6-Minute Walk Test (6MWT), and (4) able to provide written informed consent. Excluded are patient with: (1) emergency operation for major abdominal surgery, (2) medically unstable or severe comorbidities that contraindicate exercise (e.g., unstable cardiovascular disease), (3) impaired cognition or confusion, and (4) impaired vision, cognition or physical impairment of upper and lower limbs in functional task components.

**Informed consent.** Eligible patients who fulfil the inclusion criteria will be approached by the research team and invited to participate in the study. They will be provided with an information sheet outlining the study objectives, procedures, and expected time commitment together with the informed consent form prior to enrolment into the study. Upon obtaining written informed consent, participants will be formally enrolled in the study. Participants retain the autonomy to withdraw from the study at any point. Furthermore, if the principal investigator assesses that continued participation poses significant risk or harm to the participant, they will be withdrawn from the study to ensure their safety.

### Sample size estimation and justification

The sample size was calculated based on the primary outcome measure (6MWT). A recent systematic review of prehabilitation in patients undergoing major abdominal surgery reported significant improvement in 6MWT, with a mean difference of 29.4 meters (95% CI: 5.6 to 53.3 meters; p = 0.02) compared to standard care [31]. The estimated standard deviation

| | STUDY PERIOD | | | | | | |
|---|---|---|---|---|---|---|---|
| | Enrolment & allocation | Baseline | Intervention | Post-intervention assessment | Follow up | Follow up | Follow up |
| **TIMEPOINT** | February 2025- July 2026 | 3-5 week before operation | 2- 4 weeks | Preoperatively | Post-operatively prior to discharge | 4-6 weeks post-operatively | 3 months post operatively |
| Eligibility screen | X | | | | | | |
| Informed consent | X | | | | | | |
| Allocation | X | | | | | | |
| **INTERVENTIONS** | | | | | | | |
| Intervention A (Interventional) | | | X | | | | |
| Intervention B (control) | | | X | | | | |
| **ASSESSMENT** | | | | | | | |
| **Sosiodemography** | | | | | | | |
| Age | | X | | | | | |
| gender | | X | | | | | |
| Education | | X | | | | | |
| Occupation | | X | | | | | |
| **Primary Measure** | | | | | | | |
| 6 Minutes Walking Test (6MWD) | | X | | X | X | X | X |
| Maximal Inspiratory Pressure (MIP) | | X | | X | X | X | X |
| **Secondary Measure** | | | | | | | |
| Short Physical Performance Battery (SPPB) | | X | | X | X | X | X |
| 1 Minute Sit to Stand (1MSTS) | | | | X | X | X | X |
| Hand grip Test | | X | | X | X | X | X |
| Bio-impedance Test | | X | | X | X | X | X |
| Fatigue Severity Scale (FSS) | | X | | X | X | X | X |
| the Hospital Anxiety Depression Scale (HADS) | | X | | X | X | X | X |
| Euro QoL (EQ-5D-5L) | | X | | X | X | X | X |
| Self-Administered Physical Activity Questionnaire (SAQ) | | X | | X | X | X | X |
| Global Rating of Change (GRC) | | X | | X | X | X | X |

**Fig 1. SPIRIT flow diagram for the schedule of enrolment, interventions, and assessments.**

was approximately 101.6 meters, corresponding to an effect size (Cohen's *d*) of 0.29. Using this effect size, with α of 0.05 and power of 80%, the required sample size was calculated to be 54 participants. Allowing for a 30% dropout rate, a total of 70 participants (35 per group) will be recruited. Sample size calculation was performed using G*Power version 3.1.9.7.

## Procedure

**Randomisation and allocation.** Following recruitment and baseline assessments, participants will be randomly assigned to one of two groups which will be control group (MICT) or the intervention group (HIIT) using a

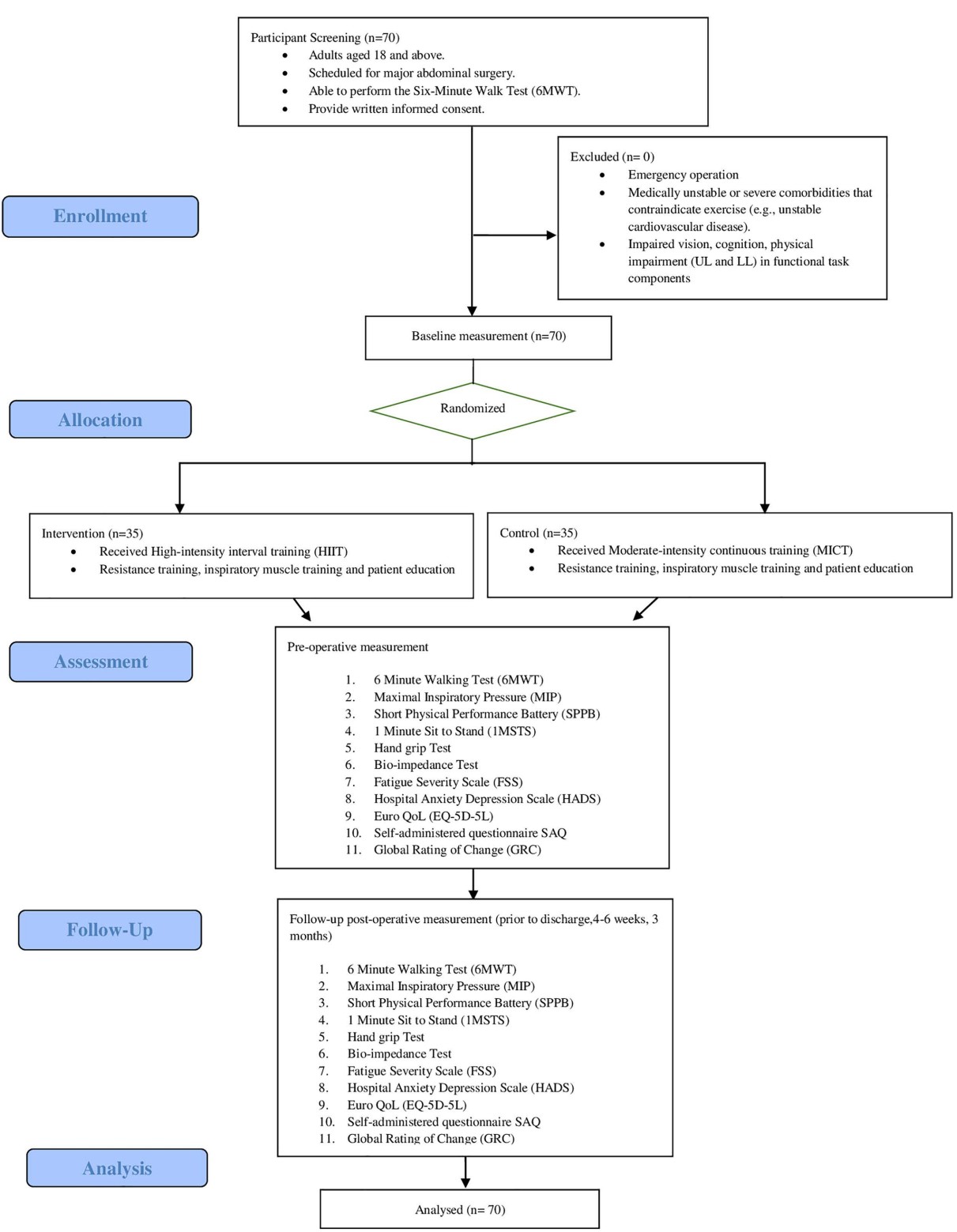

**Fig 2. Consort diagram of the study.**

computer-generated random number sequence. Allocation will be conducted by an independent individual utilizing a block randomization method (5 blocks of 14). An administrative assistant, independent of the trial, will prepare 70 sequentially numbered opaque envelopes, each containing a group allocation. A physiotherapist, trained in the study protocol, will deliver the intervention. Participants in the control group will receive moderate-intensity continuous training (MICT), while those in the intervention group will engage in high-intensity interval training (HIIT) preoperatively. Both groups will commence and conclude sessions with a warm-up and cool down to mitigate the risk of adverse events.

**Blinding.** Patients and outcome assessors, will remain blinded to treatment allocation. However, the treating physiotherapists cannot be blinded due to the nature of the intervention. To maintain blinding, details of the intervention will not be recorded in the medical records. Outcome assessments will be conducted by a blinded assessor based off-site from the outpatient department, ensuring impartiality in both inpatient and outpatient settings. Intervention sessions will occur on different days from those scheduled for outcome assessments to further reduce the risk of unblinding. If a participant from the intervention group discloses details of their intervention to the assessor, this will be documented and reported, with the rationale recorded. All analyses will be performed on an intention-to-treat basis once randomization is unblinded.

**Trial intervention.** The intervention will be delivered by the same physiotherapists at outpatient setting, based on the group allocation. All physiotherapists involved are senior clinicians with more than 5 years of experience in abdominal surgery. To ensure uniformity, the physiotherapists will have a standardized training in treating the patients. The prehabilitation exercise program is design based on recommendations from the current ERAS guidelines for major abdominal surgery [18,32]. Specifically, the program adapted from previously published protocols used in preoperative populations [1,2,14,33,34] with modifications tailored to the local clinical setting and characteristics of the target patient population.

**Standard care for control group (MICT).** This will be a tailored and supervised by a single physiotherapist (more than 10 years clinical experiences in cardiorespiratory) in outpatient hospital setting. Participants will engage in an aerobic (moderate intensity continuous training), strengthening exercise and inspiratory muscles training program prior to the date of their surgery.

For aerobic exercise this will be a low intensity in week one with graduation to moderate intensity. Moderate intensity continuous training (MICT) will allow 55–75% of maximum heart rate (MHR) [35]. Aerobic exercise training (e.g.,: treadmill, cycling, walking exercise) 1–3 times per week (60 minutes) for overall duration typical 2–4 week prior to date of surgery. Patients' Maximum Heart rate (210-age) and Karvonen Formula will be used for targeted heart rate. Patients' heart rate will be monitored along the aerobic training using smart watch. However, the Karvonen formula may not be suitable for all patient populations, particularly those on beta-blockers or with underlying cardiac conditions. In such cases, training intensity will be guided by the patient's Rating of Perceived Exertion (RPE) of 10–13 RPE score, and appropriate modifications will be made to ensure safety. This approach allows for individualized intensity adjustment while maintaining the safety and effectiveness of the intervention.

For the progressive resistance exercise, the following muscle groups will be targeted: biceps brachii, triceps brachii, pectoralis major and minor, latissimus dorsi, deltoid and rhomboids using free weights. The intensity will be 70–80% of estimated 10 repetitions maximum. The load not provided joint pain or severe muscle fatigue and/or arm fatigue on RPE of 10–13 on 6–20 RPE score [34]. Aim for a total session duration of about 45–60 minutes, including warm-up and cool-down. This includes 30–45 minutes of MICT, depending on the patient's tolerance. Sessions will be done 1–3 times per week [34].

In addition to the PVC weighted bar, we also used different types of resistance training such as band, dumbbells, kettlebells. Inspiratory muscles training will be given 20 minutes per day with 30% of maximum inspiratory pressure (MIP) [14].

Participants will also be given an illustrated handout to assist further training support by the carer (advice regarding maintaining a set amount of aerobic/strength exercise per week standard [1,2,14]. Detailed session attendance checklist and vital sign will be recorded for every attendance for adherence.

**Intervention group (HIIT).** Participants in the intervention group will receive the same care as the standard care group. Additionally, for aerobic exercise this will be a moderate intensity which is 55–75% of maximum heart rate (MHR) in week one, with graduation to high intensity in week two on > 80% of MHR [24]. Aerobic exercise training (e.g.,: Treadmill, Cycle ergometer, body weight exercises) 1–3 times per week (30 minutes) for an overall duration typically 2–4 weeks prior to the date of surgery. Patients' Maximum Heart rate (210-age) and Karvonen Formula will be used for the targeted heart rate. Patients' heart rate will be monitored along the aerobic training using a smartwatch. However, those patients who are on beta-blockers or with underlying cardiac conditions, training intensity will be guided by the RPE between 16–18 RPE score with appropriate adjustments made to ensure safety. The interval training will be 30–60 seconds of high-intensity training with 1–2 minutes of active recovery (light activity or complete rest) between intervals to facilitate recovery. This will be repeated for a total of 5–10 cycles [33].

In addition to aerobic training, participants will receive the same progressive resistance exercise and inspiratory muscle training, with intensity matched to that of the standard care of MICT group. Aim for a total session duration of about 20–30 minutes, including warm-up and cool-down. This includes 10–15 minutes of HIIT, depending on the patient's tolerance. Sessions will be done 1–3 times per week, with rest days in between to allow for recovery. Participants will also receive an illustrated handout to guide continued training support by caregivers, including recommendations on maintaining a standard amount of aerobic and strength exercise per week. A detailed session attendance checklist and vital signs will be recorded at each session to monitor adherence.

All other aspects of patient care such as preoperative management, general anaesthesia, intraoperative ventilation settings, fluid administration, prophylactic antibiotic use, pain management, management of lines and drains, general nursing care, and discharge planning will be administered at the discretion of nurses and physicians in accordance with routine clinical practices at hospital for both groups.

## Primary outcome measures

**The 6-Minute Walk Test (6MWT).** The 6MWT is a widely used measure to assess the submaximal level of functional capacity [36]. The test involves walking back and forth along a 30-meter indoor track for as many laps as possible within a 6-minute period [37,38]. Following the American Thoracic Society (ATS) guidelines, the test will be conducted before initiating prehabilitation, with the better result from two attempts recorded as the baseline measurement [39]. Participants will be informed of the elapsed time at the end of each minute during the test, but no additional encouragement will be provided [39].

**Maximal inspiratory pressure.** Inspiratory muscle strength will be assessed by measuring the maximum inspiratory pressure (MIP) generated at the mouth using the PowerBreathe K2 device. Patients will be seated in an upright position and instructed to exhale fully to their residual volume before performing a maximal and rapid inspiratory effort lasting 3 seconds. The highest 1-second average value from each trial will be recorded. To ensure reliability, three consistent measurements (defined as having a coefficient of variation <10%) will be selected from 5 to 8 maximal trials, with each trial separated by a 60-second rest interval [40].

## Secondary outcome measures

**1 Minute Sit to Stand Test (1MSTST).** The 1MSTST is recognized as a tool for assessing functional status and predicting fall risk. It has shown a good correlation with exercise capacity in patients with chronic obstructive pulmonary disease (COPD) [41]. Studies have demonstrated a significant correlation between the number of repetitions completed in the 1MSTST and essential functional outcomes in patients with COPD, including the 6-minute walk distance (6MWD), quadriceps muscle strength, and physical activity levels [41–43]. These findings suggest that the 1MSTST may serve as a valid measure of exercise capacity following respiratory rehabilitation, especially in populations with comorbid of pulmonary complications, which is common in patients with major abdominal surgery.

   

**Short Physical Performance Battery (SPPB).**  The SPPB will be used to assess lower extremity physical performance status. It is an objective outcome measures of balance, lower extremity strength, and functional capacity which includes three domains (walking, sit-to-stand and balance) [44]. SPPB had an excellent interrater reliability (ICC = 0.92) among people with respiratory condition [45]. Besides, it may serve as a reliable surrogate endpoint for all-cause mortality in clinical trials aiming to evaluate the efficacy of specific treatments or rehabilitation programs in improving health outcomes [46].

**The hospital anxiety and depression scale.**  Anxiety and depression will be measured using the 14-item Hospital Anxiety and Depression Scale (HADS), which comprises two subscales: one for anxiety (HADS-A) and one for depression (HADS-D), each containing seven items [47]. Each item is scored on a 4-point Likert scale ranging from 0 (not present) to 3 (maximum), with a maximum subscale score of 21 points [48]. A cut-off score of ≥8 points will be applied to identify cases of anxiety or depression, as this threshold has been shown to optimize sensitivity and specificity for both subscales [48]. The HADS is a validated and reliable instrument, with Cronbach's α values of 0.81 for HADS-A and 0.88 for HADS-D [48,49]. It has been widely translated and extensively utilized in international research [48,50], demonstrating high internal consistency in inpatient populations, including those undergoing median sternotomy [50].

**Hand grip strength.**  Hand grip strength will be measured in kilogram using a hand-held dynamometer with three attempts to assess physical fitness that related to muscular strength. The peak pressure of each 5-second attempt and the highest value will be recorded [51]. This hand-held dynamometer has excellent reliability (ICC = 0.98) and construct validity of $r = 0.99$ (very high correlation) with Rolyan dynamometers [52].

**EuroQol (EQ-5D-5L).**  Health related quality of life will be evaluated using self-assessment questionnaire EuroQol EQ-5D-5L. It consisted of five component scales, namely mobility, self-care, usual activities, pain, and anxiety. Recent evidence suggested that the newer 5-level version of the EQ-5D had improved measurement properties, including feasibility, ceiling effects, sensitivity, and convergent validity [53]. The EQ-5D-5L had good test-retest reliability, as indicated by intraclass correlation coefficients (ICCs) ranging from 0.65 to 0.91 for the EQ-5D index [54].

**Self-assessment of physical activity questionnaire (SAQ).**  Self-Assessment of Physical Activity Questionnaire (SAQ) will be used for physical capacity. The SAQ is a 13-item, self-administered activity questionnaire assessing common physical activities associated with personal care, ambulation, household tasks and recreation [55]. All these activities have a known metabolic equivalent (MET) obtained from the Compendium of Physical Activities. Each question required a 'yes' or 'no' response. A SAQ score will be obtained by recording the corresponding MET value of the most demanding activity that the participant perceived they could complete without symptoms [55]. All scores will be correlated against measured of peak oxygen consumption [55].

**Fatigue severity scale (FSS).**  The Fatigue Severity Scale (FSS) is a method of evaluating the impact of fatigue. The Fatigue Severity Scale (FSS) is widely used to assess the impact of fatigue on daily functioning. Its psychometric properties have been evaluated in various populations, demonstrating its reliability and validity [56]. The FSS is a short questionnaire that requires patients to rate their level of fatigue. The FSS questionnaire contains nine statements that rate the severity of fatigue symptoms. Read each statement and circle a number from 1 to 7, based on how accurately it reflects the condition during the past week and the extent to which patients agree or disagree that the statement applies to patients [57].

**The global rating of change scale.**  This is a self-reported measure of patients perceived change. It will be administered prior to the performance-based assessments preoperatively, prior to discharge, 4 weeks and 3 months postoperatively. Participants will be asked: 'Overall, how do your arms function now, compared with how your arms functioned at the initial assessment before surgery?' Responses will be recorded according to a 7-point scale from 'very much improved' to 'very much worse'. It has been previously reported that when participants rate their change as 'minimally improved', 'no change' or 'minimally worse', it is unlikely that a clinically important difference has occurred; therefore, these patients are grouped into an 'unchanged' category [58]. Responses of 'much worse', 'very much worse',

'much improved' and 'very much improved' indicate a clinically important difference has occurred and therefore these patients are grouped into a 'changed' category [58].

**Bio-electrical impedance analysis.** An InBody device utilizing Bio-electrical Impedance Analysis will be used to assess body composition by estimating key components such as lean body mass, fat mass, and total body water. This method involves passing a low-level electrical current through the body and measuring the resulting impedance to estimate body composition parameters, including total body water, extracellular water, intracellular water, fat mass, fat-free mass, body fat percentage, and bone mineral content [59].

Bioelectrical Impedance Analysis has been validated as a reliable method for assessing body composition, showing strong correlations with CT-derived measurements of torso fat volume and waist fat area, with correlation coefficients ranged from 0.86 to 0.95 in both men and women (p < 0.001) [60]. InBody devices showed a 98% correlation with dual-energy X-ray absorptiometry (DXA), with strong agreement across all body mass index (BMI) categories [61].

## Adherence monitoring

Data related to exercise prescription, progression, complaints, injuries, and symptoms during training sessions for the both groups will be systematically recorded. Participants will be encouraged to adhere to the exercise guidelines outlined in their weekly flyers. Attendance will be documented, including the total number of sessions attended and successfully completed. Detailed reasons for incomplete sessions or participant dropouts will be recorded to evaluate non-compliance. Adherence to the prehabilitation exercise program will be closely monitored, alongside compliance with the prescribed exercise regimen. Additionally, participants will be asked to report the duration of their adherence and rate their adherence to home-based activities using a numerical scale [34].

## Data collection

Demographic data and preoperative, intraoperative, and postoperative variables will be collected from participants and their medical records. Baseline assessments will be conducted 4 (± 1) weeks in outpatient setting following referral for exercise based prehabilitation prior to date of surgery. Follow up assessment will continue to 2 days (±1) before operation in the inpatient setting, prior to discharge, 4 -weeks, and 12-week post operatively. These follow-ups will take place in the research room within the physiotherapy unit.

All measurements will be conducted by an independent, trained assessor located off-site, who will remain blinded to group allocation. Face-to-face administration of follow-up tests and questionnaires will be performed by outcome assessors at four time points: preoperatively, prior to discharge, 4 weeks, and 3 months postoperatively, to maintain consistency across participants.

Post-hospital discharge follow-ups will be contacted via phone. Participants who cannot be reached by phone for 14 consecutive days from the scheduled assessment date will be considered lost to follow-up for post-discharge outcomes.

## Duration and timeline

The recruitment of all 70 participants is expected to be completed by July 2026, with data collection concluding by October 2026. The study results are anticipated in November 2026, and manuscript preparation is projected for completion by December 2026. The final manuscript will be prepared following the CONSORT extension guidelines for pragmatic trials involving non-pharmacological interventions, as outlined in Fig 1.

## Statistical analysis

The analysis will follow the intention-to-treat, whereby all participants will be analyzed according to the groups to which they were originally randomized, regardless of their adherence to the intervention protocol. SPSS version 27.0 will be used to analyse all collected data. Missing data will be handled using multiple imputation, which creates several

datasets by statistically estimating missing values and then combines the results to account for uncertainty. This method is preferred over last observation carried forward, as it reduces bias and more accurately reflects variability in clinical studies [62].

Descriptive statistics will summarize participants' baseline demographic and clinical characteristics. Continuous variables will be reported as mean (standard deviation, SD) or median (interquartile range, IQR) depending on the distribution, while categorical variables will be reported as frequency (percentage). Baseline comparability analysis of the clinical outcomes between groups will be performed using independent *t*-tests for continuous variables and Chi-square tests for categorical variables.

The primary outcomes (6MWT and MIP) and secondary outcomes (1MSTS, SPPB, hand grip strength, FSS, Bio-impedance Test, EuroQol, SAQ and Global Rating of Change Scale) will be measured at multiple time points: baseline, post-prehabilitation (4 weeks), before hospital discharge, and at 4 and 12 weeks postoperatively. To evaluate changes over time and between-group differences, a linear mixed-effects model (mixed model ANOVA) will be employed, provided the assumptions of normality and sphericity are met. Normality will be evaluated using the Kolmogorov–Smirnov test, while sphericity will be assessed using Mauchly's test. Mixed model ANOVA will provide results for time effect, group effect, and time-group interaction effect. Statistically significant time effect and group effect indicates significant within-group changes and between-group difference respectively, while significant time-group interaction effect determines if the interventions yield the desired effect on the dependent variables. Post hoc pairwise comparisons will be conducted using Bonferroni correction where appropriate.

The level of significance is set at $p < 0.05$. Effect sizes will be determined using Cohen's d, with values of 0.2, 0.5, and 0.8 indicating small, medium, and large effects, respectively [63]. Additional logistic regression will be used to determine preoperative, perioperative and postoperative risk factors associated with the development of respiratory complications. This will be an exploratory analysis, which may identify trends of predictors reported in the literature having an individual effect on postoperative respiratory complications. For all tests conducted, a p value of <0.05 (two-sided) will be considered statistically significant, and mean differences (95% CI) will be reported. The results of this study will be reported in accordance with the CONSORT 2010 guidelines.

### Data management, quality and dissemination policy

Data will be managed using an Excel database and SPSS version 27.0. Thorough training will be provided to all personnel involved in data collection, entry, and verification to ensure accuracy and consistency across measurements and to identify missing data. Regular checks will be performed to detect any inconsistencies within and between datasets. Data entry, security, and storage will follow strict protocols to ensure quality and confidentiality. All data will be encrypted and stored in a password-protected system with limited access, complying with privacy regulations. Additionally, two independent clinical members will serve in an advisory role to the clinical investigators, overseeing participant withdrawals, monitoring ethical conduct, and reviewing any serious adverse events.

Access to the final trial dataset will be limited to the principal investigator and designated members of the research team. There are no contractual agreements restricting the investigators' access to the data. This ensures the research team's full independence in data management, analysis, and reporting. The complete trial protocol is publicly available via the ANZCTR registry, and the raw dataset will be made available upon reasonable request and approval by the principal investigator. Process evaluation findings will be published in peer-reviewed journals and presented at relevant academic conferences. This study guarantees that any data presented at conferences will always maintain anonymity.

### Discussion

This study aims to fill critical gaps in the literature by evaluating the comparative effectiveness of high-intensity interval training (HIIT) and moderate-intensity continuous training (MICT) in prehabilitation among patients undergoing upper

abdominal surgery. If successful, this research could provide robust evidence supporting the use of HIIT as a time-efficient and effective intervention to enhance cardiorespiratory fitness, muscle mass preservation, and postoperative recovery. Besides that, it is offering a framework for integrating HIIT into standard prehabilitation programs, potentially transforming perioperative care practices. This study builds on prior research demonstrating the benefits of prehabilitation in improving surgical outcomes. It extends the current understanding by comparing HIIT and MICT, particularly in their effects on functional capacity, and quality of life. Previous studies, such as those by Gillen et al. (2016) and Weston et al. (2014), suggest HIIT's superiority in improving cardiorespiratory fitness [28,64]. This research will validate these findings in a surgical population, addressing a key gap in prehabilitation literature.

Several challenges may arise during the implementation of this study such as participant adherence and safety. Engaging patients in a HIIT program may require addressing barriers such as fear of exercise intensity and time constraints. HIIT, although beneficial, poses potential risks for patients with comorbidities. Careful monitoring and individualized adjustments will be necessary. Any potential adverse events during exercise sessions, including musculoskeletal strain, cardiovascular incidents, or respiratory distress will be carefully monitored in both groups. Adverse events will be systematically documented and reported. All exercise sessions will be supervised by experienced physiotherapists, and participants will be screened for exercise contraindications prior to each session to ensure safety.

If the study demonstrates HIIT's efficacy, it could lead to a paradigm shift in prehabilitation strategies for upper abdominal surgery. HIIT may not only improve patient outcomes but also optimize resource use by reducing the length of hospital stays and postoperative complications. Furthermore, establishing evidence-based exercise protocols will empower physiotherapists to play a more pivotal role in the ERAS framework.

This protocol provides a foundation for future investigations into the broader implementation of HIIT in diverse surgical populations and healthcare settings. Further studies could explore long-term outcomes, cost-effectiveness, and the integration of digital health tools to enhance adherence and monitoring.

## Supporting information

**S1 File. SPIRIT checklist.**
(PDF)

**S2 File. Ethical approval proposal.**
(PDF)

## Acknowledgments

The authors would like to thank Nurdiyana Ismail, Zamsuril Lok, Siti 'Aisyah Amran, Nur Fatihah Ahmad Yazid, Aimi Munirah Ab Rahman, for their support and contribution to the trial. They also wish to thank the physiotherapy department managers and staff at the Hospital Canselor Tuanku Muhkriz (HCTM) and Heart and Lung Centre, HCTM.

## Author contributions

**Conceptualization:** Suriah Ahmad, Nor Azura Azmi, Md Ali Katijjahbe.

**Data curation:** Suriah Ahmad, Nor Azura Azmi, Syarifah Noor Nazihah Sayed Masri.

**Formal analysis:** Suriah Ahmad, Nor Azura Azmi, Syarifah Noor Nazihah Sayed Masri, Adzim Poh Yuen Wen.

**Investigation:** Suriah Ahmad, Nur Ayub Md Ali, Md Ali Katijjahbe, Ian Chik, Adzim Poh Yuen Wen.

**Methodology:** Suriah Ahmad, Nur Ayub Md Ali, Ian Chik, Syarifah Noor Nazihah Sayed Masri, Adzim Poh Yuen Wen.

**Supervision:** Nor Azura Azmi, Nur Ayub Md Ali, Md Ali Katijjahbe, Ian Chik, Syarifah Noor Nazihah Sayed Masri.

**Validation:** Md Ali Katijjahbe.

**Visualization:** Nor Azura Azmi.

**Writing – original draft:** Suriah Ahmad.

**Writing – review & editing:** Nor Azura Azmi, Nur Ayub Md Ali, Md Ali Katijjahbe.

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
