## [Decision Letter · Decision Letter 0]

21 May 2025

Dear Dr. Azmi,

Thank you for submitting your manuscript to PLOS ONE. After careful consideration, we feel that it has merit but does not fully meet PLOS ONE’s publication criteria as it currently stands. Therefore, we invite you to submit a revised version of the manuscript that addresses the points raised during the review process.

We look forward to receiving your revised manuscript.

Kind regards,

Frederick Hong-Xiang Koh, MBBS, FRCSEd, PhD

Academic Editor

PLOS ONE

Journal Requirements:

3. Please remove your figures from within your manuscript file, leaving only the individual TIFF/EPS image files, uploaded separately. These will be automatically included in the reviewers’ PDF.

4. Please upload a copy of your study protocol that was approved by your ethics committee/IRB as a Supporting Information file. By the study protocol, we mean the complete and detailed plan for the conduct and analysis of the trial approved by the ethics committee/IRB. Please send this in the original language. If this is in a language other than English, please also provide a translation. [https://journals.plos.org/plosone/s/submission-guidelines#loc-guidelines-for-specific-study-types

Additional Editor Comments:

Please address the statistical plan and comments from the 2 reviewers before our reconsideration.

Reviewers' comments:

Reviewer's Responses to Questions

**Comments to the Author**

1. Does the manuscript provide a valid rationale for the proposed study, with clearly identified and justified research questions?

Reviewer #1: Yes

Reviewer #2: Yes

2. Is the protocol technically sound and planned in a manner that will lead to a meaningful outcome and allow testing the stated hypotheses?

Reviewer #1: No

Reviewer #2: Partly

3. Is the methodology feasible and described in sufficient detail to allow the work to be replicable?

Reviewer #1: No

Reviewer #2: Yes

4. Have the authors described where all data underlying the findings will be made available when the study is complete?

Reviewer #1: No

Reviewer #2: Yes

5. Is the manuscript presented in an intelligible fashion and written in standard English?

Reviewer #1: Yes

Reviewer #2: Yes

You may also provide optional suggestions and comments to authors that they might find helpful in planning their study.

Reviewer #1: This study protocol intended to study the prehabilitation effectiveness for patients with upper abdominal surgery. A two-armed randomized controlled trial was designed to compare high versus moderate intensity training programs with a one to one ratio. The primary outcomes are the 6-minute walk test and maximal inspiratory pressure. The secondary outcomes include 1 minute sit to stand, short physical performance battery, the hospital anxiety and depression scale, hand grip strength, euroQol, self-assessment of physical activity questionnaire, fatigue severity scale etc. Multiple visit times will be investigated and data will be in a follow-up format.

I think your sample size calculation is currently wrong. We usually do not use type I error rate of 0.9 in a sample size calculation. In addition to effect size of 30 meters difference in 6MWT, you also need to have a sensible value for the standard deviation. The results of only requiring 46 subjects probably are based on the assumption that you have very strong effect and very low variability.

The statistical analysis is a typical longitudinal data analysis plan and looks straightforward to me.

Change “two-tailed” to “two-sided” on page 19.

Are there any potential side effects that you need to discuss?

Reviewer #2: Thank you for your submission, it is a meaningful piece of work, and the results will make an impact on consideration of pre-habilitation exercise parameters for clinicians. These are points for consideration:

1) The max HR formula/ Karvonen method of calculation may not be suitable for all patient groups, there are other formulas which may be more suitable and considers past medical history such as if there are on beta blockers or not. This concerns safety, and is a necessary consideration

2) Other outcome measures such as 10m walk test ie walking speed can be considered. And also 1 minute STS vs 30s STS vs FTSTS has different advantages. Unless the patients have COPD?

3) Noted that PRE is a component of the exercise programme but there are no components of strength in the pre and post outcome measurement

4) May I check: Where is this exercise programme adopted from?

5) This exercise programme has multi components- including PRE and IMT, but the only difference between the the 2 groups are the aerobic component? Please clearly state under the intervention group, that they will also be receiving the same PRE and IMT training if so.

**Do you want your identity to be public for this peer review?** For information about this choice, including consent withdrawal, please see our Privacy Policy

Reviewer #1: No

Reviewer #2: **Yes: ** Mah Shi Min

---

## [Author Response · Author response to Decision Letter 1]

11 Aug 2025

We would like to thank the editor and the reviewers for the constructive feedback provided on our original submission. We have carefully addressed all comments and suggestions. We believe the revisions have significantly improved the clarity, scientific rigor, and presentation of the manuscript. A detailed, point-by-point response to each comment is provided in the table response to the reviewers below, along with marked and clean versions of the revised manuscript.

In addition to addressing the reviewers’ comments, we would like to inform you of several amendments to the study protocol that were made prior to and independent of the reviewers’ feedback, and which have been approved by our institutional ethics committee. Firstly, we have revised our study population from patients undergoing “upper abdominal surgery” to those undergoing “major abdominal surgery”. This change was made primarily due to improved accessibility to an adequate sample size, which is crucial for ensuring the statistical power and generalizability of our findings. Additionally, the broader classification of major abdominal surgery still encompasses upper abdominal procedures and allows for a more inclusive representation of surgical patients who may benefit from prehabilitation. This adjustment aligns with similar protocols in existing literature and supports the feasibility of recruitment within the study timeframe. This amendment to the study population has been approved by the institutional ethics committee (please refer to the letter of ethical approval of the proposal amendment, 24th June 2025).

Secondly, we also revised the study’s sample size from n= 60 to n=70, a decision made prior to receiving reviewer feedback. Additional amendments include:

i) Removal of the inclusion criterion requiring participants to achieve a 6-minute walk distance of at least 300 m, due to the limited number of eligible patients and the absence of this requirement in previous related studies;

ii) The age range for participants has been updated from 18-75 years to 18 years and above to broaden the recruitment pool,

iii) Removal of the Hospital Anxiety and Depression Scale (HADS) as a screening tool, as HADS will be used as one of the outcome measures in this study.

These amendments have been reviewed and approved by the institutional ethics committee (please refer to the letter of ethical approval of the proposal amendment, 6th August 2025).

Thank you.

---

## [Decision Letter · Decision Letter 1]

1 Sep 2025

The effectiveness of high-intensity interval training versus moderate intensity continuous training in prehabilitation among patients undergoing major abdominal surgery:  A study protocol

PONE-D-25-08010R1

Dear Dr. Azmi,

We’re pleased to inform you that your manuscript has been judged scientifically suitable for publication and will be formally accepted for publication once it meets all outstanding technical requirements.

Kind regards,

Frederick Hong-Xiang Koh, MBBS, FRCSEd, PhD

Academic Editor

PLOS ONE

Additional Editor Comments (optional):

Reviewers' comments:

Reviewer's Responses to Questions

**Comments to the Author**

1. Does the manuscript provide a valid rationale for the proposed study, with clearly identified and justified research questions?

Reviewer #1: Yes

2. Is the protocol technically sound and planned in a manner that will lead to a meaningful outcome and allow testing the stated hypotheses?

Reviewer #1: Yes

3. Is the methodology feasible and described in sufficient detail to allow the work to be replicable?

Reviewer #1: Yes

4. Have the authors described where all data underlying the findings will be made available when the study is complete?

Reviewer #1: Yes

5. Is the manuscript presented in an intelligible fashion and written in standard English?

Reviewer #1: Yes

You may also provide optional suggestions and comments to authors that they might find helpful in planning their study.

Reviewer #1: Authors now fully addressed all my previous comments. They corrected their original sample size formula and now I see no more serious issues.

**Do you want your identity to be public for this peer review?** For information about this choice, including consent withdrawal, please see our Privacy Policy

Reviewer #1: No

---

## [Editor Report · Acceptance letter]

PONE-D-25-08010R1

PLOS ONE

Dear Dr. Azmi,

I'm pleased to inform you that your manuscript has been deemed suitable for publication in PLOS ONE. Congratulations! Your manuscript is now being handed over to our production team.

Kind regards,

on behalf of

A/Prof Frederick Hong-Xiang Koh

Academic Editor

PLOS ONE